# Early Trauma-Focused Counseling for the Prevention of Acute Coronary Syndrome-Induced Posttraumatic Stress: Social and Health Care Resources Matter

**DOI:** 10.3390/jcm11071993

**Published:** 2022-04-02

**Authors:** Roland von Känel, Rebecca E. Meister-Langraf, Jürgen Barth, Hansjörg Znoj, Jean-Paul Schmid, Ulrich Schnyder, Mary Princip

**Affiliations:** 1Department of Consultation-Liaison Psychiatry and Psychosomatic Medicine, University Hospital Zurich, University of Zurich, 8091 Zurich, Switzerland; rebecca.langraf-meister@usz.ch (R.E.M.-L.); mary.princip@usz.ch (M.P.); 2Clienia Schlössli AG, 8618 Oetwil am See, Switzerland; 3Institute for Complementary and Integrative Medicine, University Hospital Zurich, University of Zurich, 8091 Zurich, Switzerland; juergen.barth@usz.ch; 4Department of Health Psychology and Behavioral Medicine, University of Bern, 3012 Bern, Switzerland; hansjoerg.znoj@unibe.ch; 5Department of Internal Medicine and Cardiology, Clinic Gais AG, 9056 Gais, Switzerland; jean-paul.schmid@kliniken-valens.ch; 6Medical Faculty, University of Zurich, 8006 Zurich, Switzerland; ulrich.schnyder@access.uzh.ch

**Keywords:** cardiovascular disease, counseling, longitudinal study, posttraumatic stress disorder, psychological stress, social support, trial

## Abstract

Background: A one-size-fits-all approach might explain why early psychological interventions are largely ineffective in preventing the development of posttraumatic stress disorder (PTSD) symptoms triggered by acute medical events. We examined the hypothesis that social and health care resources are moderators of an intervention effect. Methods: Within 48 h of hospital admission, 129 patients (mean age 58 years, 83% men) with acute coronary syndrome (ACS) self-rated their social support and were randomized to one single session of trauma-focused counseling (TFC) or stress-focused counseling (SFC) (active control intervention). Clinician-rated PTSD symptoms, use of cardiac rehabilitation (CR) and use of psychotherapy were assessed at 3 and 12 months. Random mixed regression multivariable models were used to analyze associations with PTSD symptoms over time. Results: TFC did not prevent ACS-induced PTSD symptom onset better than SFC; yet, there were significant and independent interactions between “intervention” (TFC or SFC) and social support (*p* = 0.013) and between “intervention” and duration of CR in weeks (*p* = 0.034). Patients with greater social support or longer participation in CR had fewer PTSD symptoms in the TFC group compared with the SFC group. The number of psychotherapy sessions did not moderate the intervention effect. Conclusions: Early psychological intervention after ACS with a trauma-focused approach to prevent the development of PTSD symptoms may be beneficial for patients who perceive high social support or participate in CR for several weeks.

## 1. Introduction

Post-traumatic stress disorder (PTSD) is a debilitating mental disorder that can develop, at the earliest, one month after a traumatic event [1], including life-threatening diseases [1] such as acute coronary syndromes (ACS) [2,3]. About 12% of patients have clinically relevant PTSD symptoms after ACS [4] such as reliving aspects of the cardiac event in thoughts or dreams, avoiding reminders cued to aspects of the event, and increased arousal [5]. ACS-induced PTSD symptoms are associated with poor quality of life [6] and increased risk of recurrent cardiovascular events and mortality [4,5]. For these reasons, and other adverse consequences for physical and mental health, offsetting the risk for PTSD symptoms triggered by ACS through preventive measures are of clinical importance [3]. We previously conducted the Myocardial Infarction-Stress Prevention Intervention (MI-SPRINT) trial with a psychological first aid approach for patients with a high level of distress during ACS [7]. However, we were unable to demonstrate a benefit of early trauma-focused counseling (TFC) vs. stress-focused counseling (SFC), as an active control, in terms of reduced ACS-induced PTSD symptoms at 3 and 12 months [8,9]. This sobering finding is consistent with a recent meta-analysis of randomized controlled trials showing little evidence that early psychological interventions are effective in preventing the onset of PTSD symptoms after life-threatening medical events [10].

To advance this area, it was suggested that future research should examine how the effectiveness of interventions varies among individuals who become acutely traumatized by a medical event [10]. Similarly, the larger trauma literature has emphasized the need to develop treatments that promote tailoring of interventions to a patient’s individual needs and characteristics, rather than a one-size-fits-all approach [11]. The search for intervention moderators has been particularly emphasized to identify patients most likely to benefit from both early prevention and treatment of PTSD symptoms [10,11].

Social and health care resources could be moderators of an early preventive intervention effect on ACS-induced PTSD symptoms if they are a prerequisite for patients to benefit from trauma-focused elements of the intervention and their use is taught as part of the intervention. For instance, there is a cross-sectional and longitudinal relationship between greater social support and lower self-reported PTSD symptoms in trauma-exposed non-clinical populations [12] with similar findings in patients with ACS [13,14]. Significant others might help patients to normalize symptoms after a heart attack. Based on a trusting relationship, they could encourage them to refrain from avoidance tendencies and other maladaptive coping strategies. Nevertheless, it should also be mentioned that significant others may unintentionally worsen PTSD symptoms in patients with ACS by transmitting their distress onto their loved ones [15].

In terms of health care resources, use of cardiac rehabilitation (CR) could help patients exercise despite physical arousal, which is difficult to distinguish from cardiac symptoms acting as intrusive reminders of a recurrent heart attack [16] and thus regain confidence in a functioning heart. In a sample of patients with mainly ACS, clinically relevant PTSD symptoms had decreased after several weeks of CR, but were higher in patients with more cardiac misconceptions (e.g., “people with angina should always avoid things that bring it on”) both before and after CR [17]. This may indicate an association of cognitive and emotional adjustment post-ACS and that addressing this relationship as part of TFC might result in lower PTSD symptoms. Furthermore, patients who are informed that PTSD symptoms may occur after hospitalization may recognize them earlier and follow the advice conveyed in the intervention to see a psychotherapist when they become distressed. 

Based on the empirical evidence and theoretical considerations delineated above, we explored the hypothesis that patients with ACS who had participated in MI-SPRINT would derive greater benefit from TFC than from SFC if they had resources in their social and health care environment to implement trauma-focused information in their daily lives. We specifically hypothesized that in patients who had received TFC, greater social support at hospital admission, longer duration of CR, and more psychotherapy sessions in the post-hospital period would be associated with fewer PTSD symptoms over 12 months after ACS.

## 2. Materials and Methods

### 2.1. Study Design and Participants

This is a secondary analysis of data collected as part of the MI-SPRINT randomized controlled trial [7] that enrolled 190 consecutive patients who underwent acute coronary intervention due to ACS at a university hospital in Switzerland between February 2013 and September 2015 [8,9]. The ethics committee of the State of Bern, Switzerland, approved the study protocol, which was registered under ClinicalTrials.gov (NCT01781247). All participants provided written informed consent. The primary aim of MI-SPRINT was to test the hypothesis that one 45-min session intervention of TFC, delivered within 48 h of hospital admission, prevents the development of interviewer-rated ACS-induced PTSD symptoms compared with SFC (active control intervention). As published elsewhere, TFC did not perform better than SFC at both 3-month and 12-month follow-up [8,9].

To participate in MI-SPRINT, patients had to be 18 years or older, have verified ST-segment elevation myocardial infarction (STEMI) or non-STEMI, stable circulation, and peritraumatic distress (i.e., a score of at least 5 for pain intensity plus at least 5 for fear of dying and/or feelings of helplessness on numeric rating scales ranging from 0–10) [18]. The latter was required because early psychological interventions aimed at preventing the development of PTSD after trauma exposure may be more effective than usual care only for individuals with traumatic stress symptoms [7,19]. Patients were excluded if they needed emergency cardiac surgery, had severe comorbid diseases, limited orientation, cognitive impairment, current severe depression based on the cardiologist’s clinical judgement, suicidal ideations in the last 2 weeks, insufficient knowledge of German, or were enrolled in another randomized controlled trial.

For the present analysis, we used data of a subsample of participants who underwent the clinical interview to assess ACS-induced PTSD symptoms 3 and/or 12 months after ACS and who had available information on social support assessed with the ENRICHD Social Support Inventory (ESSI) at hospital admission. Of the originally enrolled 190 participants, clinical PTSD symptoms could be assessed in 50 patients at 3-month follow-up only, in 2 at 12-month follow-up only, and in 104 at both follow-ups. Recruitment and reasons for dropouts have been described elsewhere [8,9]. Because 27 (17.3%) of the 156 participants assessed for PTSD symptoms did not complete the ESSI, we decided not to replace the missing social support data with multiple imputation, resulting in 129 participants whose data were available for the present analysis.

### 2.2. Intervention Aimed at Preventing ACS-Induced PTSD Symptoms

An independent person randomly assigned eligible participants to either one session of TFC or SFC with a computer-generated algorithm. Allocation was only accessible to researchers after the end of the entire study. Both TFC and SFC were active face-to-face interventions of the same duration and care, conducted by the bedside on the coronary care unit, within 48 h of hospital admission. Doctoral students in psychology and medicine, trained and supervised by senior clinical psychotherapists, acted as counselors. Each session lasted 45 min. Two illustrated booklets, each tailored to the intervention strategy (i.e., TFC or SFC), were used as information material for interacting with patients during the counseling session and to elaborate on certain topics. Both booklets are available as online material [8] and served for further self-guided help after hospital discharge. Among other content, the booklets informed about the importance of support from the personal environment or through professional help for psychological adjustment after a heart attack and about psychological distress as a barrier to participate in CR. Following a psychological first aid approach, the counselor initially dealt with a patient’s most immediate concern, before addressing key topics such as “normalization of stress reactions”. For TFC, an educational approach was taken, targeting a patient’s personal resources and cognitive (re)structuring to prevent potentially occurring ACS-related traumatic reactions. As concepts, psychological trauma and PTSD were introduced, including that a heart attack may induce PTSD symptoms. In contrast, SFC conveyed the important role of psychosocial stress in heart disease in general and how to use this information for life after a heart attack. Any trauma-related terminology was strictly avoided. A more detailed description of both sessions’ content can be found elsewhere [7,8,9]. 

### 2.3. Measures

#### 2.3.1. Timing of Data Collection

At hospital admission, research assistants obtained data from patient files and through a structured medical history, and asked patients to self-rate social support with the ESSI. For the 3-month and 12-month follow-up, patients were invited to undergo a clinical interview for the assessment of ACS-induced PTSD symptoms and to self-rate depressive symptoms. At both follow-up visits, a structured medical history was taken, asking about health behaviors, participation in CR, attendance in psychotherapy, and use of informational materials, covering the past 3 and 9 months, respectively.

#### 2.3.2. PTSD Symptoms

Symptoms of PTSD at 3 and 12 months post-ACS were assessed with the validated German version of the Clinician-Administered PTSD Scale (CAPS) with reference to the Diagnostic and Statistical Manual for Mental Disorders (DSM)-IV criteria [20]. Clinician ratings of each of the 17 PTSD symptoms range from 0 to 4 for both frequency (0 = “never”, 4 = “daily or almost every day”) and intensity (0 = “none”, 4 = “extreme”) and were anchored to the index MI as the traumatic event. This results in a PTSD symptom score between 0 and 136, composed of 5 re-experiencing symptoms, 7 avoidance/numbing symptoms, and 5 hyperarousal symptoms. A CAPS score ≥20 indicates clinically significant PTSD symptoms [21]. The minimal clinically important difference on the CAPS score is 10 points [22]. Cronbach’s α for the PTSD total symptom score was 0.79 at 3 months and 0.68 at 12 months, indicating acceptable internal consistency of the scale overall.

#### 2.3.3. Personal and Medical Resources

Social support was assessed at hospital admission using a German version [23] of the ESSI [24]. The ESSI has been shown to be a valid and reliable measure of social support in patients undergoing percutaneous coronary intervention [25]. The 7-item scale covers structural (partner, 1 item), informational (advice, 1 item), instrumental (tangible help, 1 item), and emotional (caring, 4 items) support from any network member. Item examples are “Is there someone available to help you with daily chores?” and “Is there someone available to you who shows you love and affection?” Response categories range from 1 (“none of the time”) to 5 (“all of the time”), with item 7 (currently married or living with a partner) scored 4 for “yes” and 2 for “no” [24], resulting in a social support score between 8 and 34. In our study sample, Cronbach’s α was 0.85, indicating good internal consistency of the scale.

At the 3- and 12-month follow-up, participants were asked whether they had used inpatient and/or outpatient CR and whether they had used psychotherapy since hospital discharge or the 3-month assessment. If “yes,” they were also asked to indicate the duration of participation in CR in weeks and the number of psychotherapy sessions attended. As an example, a patient who started with a 12-week CR program 4 weeks after hospital discharge, used CR for 8 weeks between hospital discharge and the 3-month follow-up and for 4 weeks between the 3-month and the 12-month follow-up.

#### 2.3.4. Time-Invariant Covariates

Time-invariant covariates were assessed at hospital admission. Demographic variables included age, sex, and educational level. The latter was categorized as high (university graduation, including applied sciences/high school graduation/matura), medium (apprenticeship or vocational school), or low (lower than apprenticeship or vocational school) [26]. Index MI (STEMI or non-STEMI), the number of coronary vessels diseased (i.e., luminal narrowing ≥ 50%), and angiographic left ventricular ejection fraction (LVEF) were used as indices of ACS severity. The level of peritraumatic distress was quantified as the sum divided by three of numerical ratings for perceived pain, fear of dying, and helplessness during ACS required for inclusion in the study (see above). The Charlson comorbidity index was categorized by high, intermediate, or low 10-year mortality risk [27]. A 3-item screener was used to identify probable PTSD cases due to traumatic experiences in the 3 months prior to the index MI [28]. Depression history was assessed asking patients the question “Have you ever had a depression in your life? (yes/no)”.

#### 2.3.5. Time-Variant Covariates

Time-variant covariates were assessed at 3 and 12 months. Depressive symptoms were assessed with the 21-item Beck Depression Inventory (BDI) [29]. Participants rated the severity of each symptom with a value between 0 and 3, resulting in a total score between 0 and 63. Smoking status was categorized as either current or former/never smoker. Physical activity per week “that makes you sweat” was categorized as none, 1–2× or 3–7×. Based on epidemiologic evidence as a cardiovascular risk factor, alcohol consumption was categorized as heavy (“2”) (>21 standard drinks/week for men and >14 for women), none (“1”) or moderate (“0”). Use of information material was assessed with the question “Since hospital discharge/the 3-month assessment, how often have you consulted the information material on “stress and heart attack” that we handed to you?” Response categories were “never”, “1×/month”, “1×/week”, “>1×/week” or “daily”. Across both assessments, weekly use was mentioned only 4 times, so this variable was categorized as ≥1/month or never.

### 2.4. Statistical Analyses

Data were analyzed using SPSS 27.0 for Windows (SPSS Inc., Chicago, IL, USA) with a two-tailed significance level of *p* < 0.05. Descriptive statistics are reported as means (M) and standard deviations (SD), median and inter-quartile range (IQR) or *n* (%) as appropriate. Due to a non-normal distribution, the PTSD symptom score (dependent variable) was square root transformed prior to analysis. Only a few data were missing and handled as follows. Two participants each missed one item response of the ESSI scale, which was replaced by the mean of the completed items. Multiple imputation (k = 5) was used to replace missing data for the other time-invariant covariates, i.e., four missing values for PTSD screen and two each for LVEF, previous MI, and depression history. Of time-variant covariates, there were four missing values for BDI scores, two for psychotherapy sessions, and one each for weeks of CR and use of information material. Missing BDI scores, CR weeks, and, in one case, psychotherapy sessions were replaced with the median of the respective assessment. A “0” for psychotherapy sessions was assigned to the other participant, who had no psychotherapy in the first 3 months after ACS and lacked this information for the interval between 3 and 12 months. The participant who did not answer the question about use of informational material at 3 months was assigned the response “never” based on a “never” response for the interval between 3 and 12 months.

The longitudinal association of the “intervention” (TFC vs. SFC), available resources (social support, CR duration, psychotherapy sessions) and interactions thereof with PTSD symptoms over time (i.e., across the 3- and 12-month assessment) was examined with linear mixed (random effects) regression analysis. Covariates were selected a priori based on potential associations with PTSD symptoms (2) and treated as fixed effects. Age, sex, education, index MI, number of coronary vessels diseased, LVEF, peritraumatic distress, previous MI, comorbidity index, PTSD screen at admission, depression history, and social support were entered as time-invariant variables. Depressive symptoms, smoking status, alcohol consumption, physical activity, use of information material, weeks of CR, and number of psychotherapy sessions were entered as time-variant variables. Exploratory analyses examined whether resources-related intervention effects were modified by the use of information material after hospitalization and the time of participation in CR.

## 3. Results

### 3.1. Participant Characteristics

Table 1 shows the characteristics of the 129 study participants in each intervention group who provided data at 3 months (*n* = 127) and 12 months (*n* = 87) after ACS, resulting in 214 assessments across the 12-month observation period. The mean (SD) age of the 129 participants was 58.2 (11.2) years, 19 (14.7%) were female, and 67 (51.9%) were randomized to TFC. The distribution of the time-invariant variables collected at hospital admission, including social support, looked similar at 3 months and 12 months, suggesting no evident attrition of participants over time with respect to these characteristics. In terms of time-variant variables, use of CR and, probably related, physical activity were more frequent in the first 3 months than between 3 and 12 months after ACS. While approximately the same proportion of participants attended psychotherapy during both observation periods, patients reported using the informational material (illustrated text booklet) given to them during hospitalization more frequently during the first 3 months after ACS than thereafter.

### 3.2. PTSD Symptoms and Intervention Effect

Based on a CAPS score ≥20, the prevalence of clinically significant ACS-induced PTSD symptoms was 18.9% (*n* = 24) at 3 months and 5.7% (*n* = 5) at 12 months. However, compatible with a similar median CAPS score at both assessments (Table 1), there was no significant “time” effect (Table 2), suggesting that continuous scores of PTSD symptoms were stable over time. As previously reported [8,9], there was also no significant main effect for “intervention” (TFC or SFC) and no significant intervention-by-time interaction in both the univariable (*p* = 0.33) and the multivariable (*p* = 0.24) model, indicative of equal severity of PTSD symptoms over time in both intervention groups.

### 3.3. Effects of Social and Health Care Resources

#### 3.3.1. Main Effects of Resources

As can be seen in Table 2, a greater number of psychotherapy sessions was significantly associated with more PTSD symptoms over time in both the univariable (*p* < 0.001) and multivariable (*p* = 0.004) analysis. Greater social support was associated with fewer PTSD symptoms over time in the univariable analysis only (*p* = 0.005), and duration of CR did not show a significant association with PTSD symptoms in either analysis.

#### 3.3.2. Moderation of Intervention Effect on PTSD Symptoms by Resources

There were significant interactions between “intervention” and social support (*p* = 0.013) and between “intervention” and duration of CR (*p* = 0.034) for PTSD symptoms over time, adjusting for main effects of predictor variables and the other covariates in the multivariable model (Table 2). No significant interaction emerged between “intervention” and the number of psychotherapy sessions over time. There were also no significant interactions between “intervention” and psychotherapy (yes/no) and between “intervention” and CR (yes/no) for PTSD symptoms over time.

Follow-up analyses revealed an association between greater social support and fewer total PTSD symptoms over time in the TFC group [estimate (SE) −0.118 (0.033), *p* < 0.001] compared with the SFC group [0.024 (0.031), *p* = 0.45]. Analysis with original (i.e., not square root transformed) units of the PTSD symptom score showed a 8.6-point decrease in the TFC group and a 1.3-point increase in the SFC group with a 12-point increase in social support. This group difference of 10 corresponds to the minimal clinically important difference. Furthermore, there was an association between longer duration of CR and fewer PTSD symptoms over time in the TFC group [−0.036 (0.027), *p* = 0.20] compared with the SFC group [0.031 (0.024), *p* = 0.20]. Using the original units of the PTSD symptom score for this follow-up analysis indicated that 20 weeks of CR would be required to achieve a group difference of 10 points (4.5-point decrease in the TFC group vs. 5.4-point increase in the SFC group).

Supplementary multivariable analyses for duration of outpatient and inpatient CR separately revealed a significant interaction between “intervention” and duration of outpatient CR [−0.066 (0.031), *p* = 0.031], adjusting for duration of inpatient CR. In contrast, the interaction between “intervention” and duration of inpatient CR, adjusting for duration of outpatient CR, was not significant (*p* = 0.76).

#### 3.3.3. Exploratory Analyses

A three-way interaction between “intervention”, use of information material over time and social support showed a trend towards significance for PTSD symptoms [−0.144 (0.082), *p* = 0.082]. Follow-up analysis showed a greater decrease in PTSD symptoms over time associated with increased social support among participants in the TFC group with use of information material [−0.119 (0.039), *p* = 0.003] vs. no use [−0.066 (0.083), *p* = 0.44]. There were no such significant 3-way interactions with CR duration and psychotherapy sessions. The lack of a significant three-way interaction between “time”, “intervention”, and duration of CR suggests that the time when participants used CR after ACS was not critical to achieve a decrease in PTSD symptoms with longer duration of CR.

### 3.4. Additional Associations with PTSD Symptoms over Time

Table 2 shows that in addition to the associations of primary interest reported above, there were further independent associations of several covariates with PTSD symptoms over time. Greater peritraumatic distress during ACS (*p* < 0.001) and more depressive symptoms over time (*p* < 0.001) were both associated with more PTSD symptoms over time. More frequent physical activity was also associated with more PTSD symptoms (*p* = 0.006).

## 4. Discussion

As in our previous publications [8,9], one single session of early TFC was not more effective than SFC in preventing the development of ACS-induced PTSD symptoms in the subgroup of 129 participants in the MI-SPRINT trial studied here. However, the present study yielded important new findings suggesting that TFC may actually be more effective than SFC if patients can access resources from their social environment and the health care system. Specifically, we found that patients with ACS and higher levels of social support had fewer PTSD symptoms over 12 months if they received early TFC compared with SFC. This relationship was independent of covariates previously associated with ACS-induced PTSD symptoms [2], such as peritraumatic stress and depressive symptoms, both also predictors in our study sample. This finding may be clinically relevant because a 12-point higher social support score (range 8–34) in the TFC group was associated with a 10-point lower PTSD symptom score than in the SFC group, a difference considered clinically important [22].

The observed benefit of social support for patients who underwent early TFC is consistent with the large body of evidence showing that social support contributes to good health through various direct and indirect mechanisms [30]. Social support is defined as the perceived or actual receipt of social resources (e.g., tangible, emotional) and is a reliable predictor of physical and mental health outcomes [31,32], including mortality [33]. Particularly relevant to the context of our study, increased social support has been associated with reduced risk of incident and recurrent ACS [34] and fewer PTSD symptoms in various samples exposed to different types of trauma [12], including ACS [2]. In our study, social support assessed at hospital admission was also significantly associated with fewer PTSD symptoms over time, but in the unadjusted analysis only. Significant inverse associations between social support and ACS-induced PTSD symptoms without adjustment for covariates have also been reported in previous observational studies at 4–6 weeks [13], at 3 months [35] and between 6 and 12 months [36] after ACS. Two of these studies included covariates, including negative affect [36] and neuroticism [13], which also rendered the associations nonsignificant. Another study showed an independent association between higher social support and fewer PTSD symptoms 1 and 6 months after ACS, but did not control for cardiac disease severity, comorbidities, and health behaviors [14].

Because these largely cross-sectional studies included fewer than 150 patients each [2], prospective studies with larger samples are warranted and should also explore indirect pathways leading from higher social support to fewer ACS-induced PTSD symptoms. Our results suggest that appropriate early psychological interventions may represent such a pathway in patients who report high peritraumatic distress during ACS. A cautious interpretation of our exploratory analysis could be that patients who used the informational material about how to manage trauma-related symptoms and behaviors in the post-hospital period may increase the benefit of TFC if social support is high. One possibility could be that significant others may help counteract trauma-related behaviors through informal and natural exposure. Such an assumption aligns with a recent meta-analysis showing that psychosocial behavioral support interventions improved survival in different health care settings, but interventions that focused primarily on patients’ social or emotional outcomes did not [37].

We further found that the longer patients in the TFC group participated in CR, the fewer PTSD symptoms they had over time compared with patients in the SFC group. In accordance with recommendations of the European Society of Cardiology for patients hospitalized for an acute coronary event [38], 84.3% of our patients started with a CR program in the first 3 months after ACS. It is recommended to start CR soon after ACS. However, an exploratory analysis suggested that patients could expect fewer ACS-related PTSD symptoms during the 12-month observation period regardless of whether they started CR earlier or later. Effect estimates suggested that 20 weeks of CR would be necessary for a clinically relevant difference in PTSD symptoms between the two intervention groups. Admittedly, this is difficult to achieve in everyday practice in Switzerland, where outpatient CR takes between 6 and 12 weeks and inpatient CR between 2 and 4 weeks. Nevertheless, even at a subclinical level, PTSD symptoms have been associated with an increased risk of recurrent cardiovascular events [5]. The vast majority of study participants used outpatient CR. Therefore, the longer duration of outpatient CR and limited statistical power due to the small proportion of patients using inpatient CR, may be reasons why longer duration of outpatient CR was significantly associated with fewer PTSD symptoms whereas duration of inpatient CR was not.

In addition to a high volume of aerobic exercise training and group lectures on managing cardiovascular risk factors and psychosocial issues, some patients also use psychotherapy during CR [39]. However, the effect of CR on PTSD symptoms in the TFC group was independent of psychotherapy attendance. Thus, other components of CR allowing, for example, correction of cardiac misconceptions (16) must have been particularly effective in the TFC group. Unfortunately, we were not able to distinguish whether patients received psychotherapy in the CR setting or outside the CR setting, e.g., in a private psychotherapy practice. However, psychological support for patients attending CR is also provided by the team of cardiologists, psychiatrists, and/or psychologists. To determine exactly which elements of CR help patients who received early TFC most to manage trauma-related issues, a mixed methods study combining quantitative and qualitative data could be informative. Clearly, support from psychologists/psychiatrists and cardiologists during CR is important and may have an impact (despite TFC) in patients who have experienced an ACS, the effect of which needs further investigation.

The fact that psychotherapy sessions were significantly associated with PTSD symptoms may indicate that patients with more severe PTSD symptoms were particularly motivated to use psychotherapy. Physical exercise was also directly associated with PTSD symptoms. This could reflect the fact that both intervention groups were informed that psychotherapy and exercise are means to alleviate distress. However, the intervention did not moderate the association between psychotherapy sessions and PTSD symptoms. We can therefore speculate that TFC did not result in patients receiving trauma-focused psychotherapy, which has been shown to improve PTSD symptoms in patients with an acute cardiovascular event and high baseline PTSD symptoms [40].

The repetitive assessment of interviewer-rated PTSD symptoms and health care resources over one year was a strength of our study. Confidence in the results of the study is limited by the fact that it is a secondary and partially exploratory analysis of a subsample of patients with complete data of interest. Particularly, we did not consider adjusting our significance tests for multiple comparisons, so the significant findings should be interpreted with great caution. Moreover, the sample characteristics with a high proportion of well-educated, predominantly male participants with low somatic comorbidity limit the generalizability of study results. Because 85% of our study participants were men, it is unclear whether the results would hold true in a sample that was more gender balanced or predominantly female. It is unclear whether the moderated intervention effects could be replicated with the new DSM-5 criteria for PTSD. It should be noted that even in a health care system that provides access to CR and psychotherapy on the basis of compulsory health insurance, as is the case in Switzerland, these resources can only be beneficial if patients with ACS are willing to use them. In addition, while interventionists could potentially select interventions based on differing amounts of social support, it would be challenging to select based on CR or psychotherapy attendance as this would not be known for certain at the time of treatment selection.

## 5. Conclusions

This controlled intervention study found that one single session of early TFC, which included information about using social and health care resources, prevented ACS-induced PTSD symptoms in patients with high social support and those who participated in CR for several weeks. The findings could inform future studies of early psychological interventions after an acute medical event to consider social and health care resources of traumatized patients as moderators of treatment outcome.

## Figures and Tables

**Table 1 jcm-11-01993-t001:** Characteristics of study participants in each intervention group at the two assessments.

Variables Assessed	3-Month Follow-Up (*n* = 127)		12-Month Follow-Up (*n* = 87)	
*Time-invariant variables*	TFC (*n* = 66)	SFC (*n* = 61)	TFC (*n* = 48)	SFC (*n* = 39)
Age, years, M (SD)	59.4 (11.1)	56.2 (10.4)	59.4 (10.2)	58.8 (9.2)
Sex, female, *n* (%)	11 (16.7)	8 (13.1)	9 (18.8)	6 (15.4)
Educational level				
High, *n* (%)	19 (28.8)	8 (13.1)	13 (27.1)	5 (12.8)
Medium, *n* (%)	44 (66.7)	50 (82.0)	32 (66.7)	31 (79.5)
Low, *n* (%)	3 (4.5)	3 (4.9)	3 (6.3)	3 (7.7)
ST-elevation MI, *n* (%)	48 (72.7)	44 (72.1)	32 (66.7)	28 (71.8)
Coronary vessels diseased, M (SD)	1.88 (0.89)	1.80 (0.83)	1.83 (0.88)	1.77 (0.84)
LVEF, %, M (SD)	49.0 (11.8)	47.2 (11.3)	51.0 (11.9)	47.2 (11.7)
Peritraumatic distress, M (SD)	6.37 (1.31)	5.93 (1.36)	6.61 (1.31)	5.85 (1.44)
Previous MI, *n* (%)	3 (4.5)	6 (9.8)	2 (4.2)	5 (12.8)
Comorbidity index				
High risk, *n* (%)	12 (18.2)	6 (9.8)	9 (18.8)	5 (12.8)
Medium risk, *n* (%)	14 (21.2)	15 (24.6)	10 (20.8)	13 (33.3)
Low risk, *n* (%)	40 (60.6)	40 (65.6)	29 (60.4)	21 (53.9)
PTSD screen positive, *n* (%)	9 (13.6)	5 (8.2)	6 (12.5)	2 (5.1)
Depression history, *n* (%)	15 (22.7)	20 (32.8)	8 (16.7)	12 (30.8)
Social support, M (SD)	29.5 (4.2)	29.1 (4.9)	29.5 (4.2)	29.5 (5.2)
*Time-variant variables*				
PTSD symptoms, median (IQR)	7.5 (2.8, 17.3)	8.0 (4.0, 13.0)	8.0 (3.75, 14.0)	8.0 (5.0, 14.0)
Depressive symptoms, median (IQR)	5.0 (1.8, 8.0)	4.0 (2.0, 8.0)	5.0 (1.0, 7.0)	5.0 (2.0, 8.0)
Current smoker, *n* (%)	8 (12.1)	7 (11.5)	6 (12.5)	6 (15.4)
Alcohol consumption				
Moderate, *n* (%)	50 (75.8)	46 (75.4)	38 (79.2)	30 (76.9)
None, *n* (%)	12 (18.2)	13 (21.3)	6 (12.5)	9 (23.1)
Heavy, *n* (%)	4 (6.1)	2 (3.3)	4 (8.3)	0 (0)
Physical activity				
None, *n* (%)	13 (19.7)	11 (18.1)	11 (22.9)	9 (23.1)
1–2×/week, *n* (%)	10 (15.2),	9 (14.8)	16 (33.3)	7 (17.9)
3–7×/week, *n* (%)	43 (65.2)	41 (67.2)	21 (43.8)	23 (59.0)
Cardiac rehabilitation, *n* (%)	56 (84.8)	51 (83.6)	18 (37.5)	15 (38.4)
Outpatient CR, *n* (%)	44 (66.7)	46 (75.4)	17 (35.4)	13 (33.3)
Inpatient CR, *n* (%)	10 (15.1)	4 (6.6)	1 (2.1)	2 (5.1)
Outpatient and inpatient, *n* (%)	2 (3.0)	1 (1.6)	0 (0)	0 (0)
CR duration, weeks, mean (SD)	7.2 (4.3)	7.3 (4.4)	2.1 (5.1)	3.0 (7.1)
Outpatient CR, weeks, mean (SD)	6.6 (4.9)	7.0 (4.6)	2.0 (5.1)	2.9 (7.1)
Inpatient CR, weeks, mean (SD)	0.6 (1.4)	0.2 (0.8)	0.1 (0.4)	0.2 (0.7)
Psychotherapy, *n* (%)	16 (24.2)	16 (26.2)	9 (18.8)	10 (25.6)
Psychotherapy sessions, mean (SD)	0.6 (1.2)	1.1 (2.4)	2.1 (6.3)	1.7 (4.0)
Information material use ≥1×/month, *n* (%)	62 (93.9)	54 (88.5)	23 (47.9)	17 (43.6)

CR, cardiac rehabilitation; IQR, interquartile range; LVEF, left ventricular ejection fraction; M, mean; MI, myocardial infarction; PTSD, posttraumatic stress disorder; SD, standard deviation; SFC, stress-focused counseling; TFC, trauma-focused counseling.

**Table 2 jcm-11-01993-t002:** Relationships with PTSD symptoms over 12 months.

Parameters in Model	Univariable		Multivariable	
	Estimate	SE	Estimate	SE
Intercept	2.762 ***	0.129	2.500	1.305
Time	−0.124	0.141	−0.298	0.214
Age	−0.023	0.012	−0.004	0.010
Female sex	0.651	0.356	−0.007	0.294
Education	−0.266	0.266	−0.216	0.199
ST-segment elevation MI	0.118	0.288	0.111	0.244
Coronary vessels diseased	0.240	0.150	0.149	0.112
LVEF	−0.010	0.011	−0.005	0.010
Peritraumatic distress	0.367 ***	0.089	0.268 ***	0.074
Previous MI	−0.124	0.502	0.295	0.379
Comorbidity index	−0.120	0.176	−0.011	0.134
PTSD screen positive	−0.004	0.418	−0.426	0.324
Depression history	0.645 *	0.287	0.216	0.243
Depressive symptoms	0.147 ***	0.021	0.130 ***	0.021
Current smoker	−0.060	0.327	−0.161	0.282
Alcohol consumption	−0.058	0.178	−0.065	0.163
Physical activity	0.127	0.123	0.340 **	0.123
TFC	0.150	0.258	0.110	0.199
Use of information material	−0.164	0.187	−0.418	0.242
Social support	−0.079 **	0.028	−0.037	0.022
Rehabilitation weeks	0.022	0.015	0.003	0.017
Psychotherapy sessions	0.090 ***	0.025	0.075 **	0.026
*Additionally included interaction terms (3 separate multivariable models)*				
TFC × social support			−0.104 *	0.041
TFC × rehabilitation weeks			−0.066 *	0.031
TFC × psychotherapy sessions			−0.038	0.054

LVEF, left ventricular ejection fraction; MI, myocardial infarction; PTSD, posttraumatic stress disorder; SE, standard error; TFC; trauma-focused counseling; Total PTSD symptoms (dependent variable) were entered as square root transformed values of the Clinician-Administered PTSD Scale total score. Significance level: *** *p* < 0.001; ** *p* < 0.010; * *p* < 0.05.

## Data Availability

The anonymized data that support the findings of this study are available from the corresponding author on reasonable request.

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
