# Peer review of "Early Trauma-Focused Counseling for the Prevention of Acute Coronary Syndrome-Induced Posttraumatic Stress: Social and Health Care Resources Matter"

_jcm, 2022, doi:10.3390/jcm11071993_

Round 1
Reviewer 1 Report
This manuscript explores factors that may have influenced the effect an early trauma-focused counseling (TFC) intervention in ACS patients in a randomized clinical trial that enrolled 128 patients. The authors had previously shown that there were no differences in 3-mo or 12-mo ACS-related PTSD symptoms in patients who received the TFC intervention versus a stress-focused counseling active control. Here, they explored whether social support, cardiac rehabilitation participation, and psychotherapy sessions moderated the intervention effect. Overall, the manuscript was clearly presented and used sound statistical methods. Given the relative paucity of knowledge regarding early interventions to prevent PTSD symptoms after acute cardiovascular events, the manuscript makes an important contribution to the understanding of these interventions. The finding that social support may influence the effect of TFC, if confirmed, could be one way in which psychological interventions are personalized. There were also a few concerns with the manuscript that diminished enthusiasm. Some comments that may be useful to address are provided below:
- In the background section, the authors highlight literature showing social support could help mitigate PTSD symptoms after ACS. They could also consider literature that shows that some partners may unintentionally worsen PTSD symptoms in ACS patients by transmitting their distress onto their loved ones. (e.g., Cornelius et al. Gen Hosp Psych 2019).
- The authors utilize cardiac rehabilitation as an example of health care resources. Yet, attendance at cardiac rehabilitation is often driven as much by patient refusal or drop out rather than resource access. Was lack of attendance or lower number of cardiac rehabilitation visits ever driven by insurance/access issues that were outside the patients’ control? Further, while interventionists could potentially select interventions based on differing amounts of social support, it would be challenging to select based on cardiac rehabilitation or psychotherapy attendance as this would not be known at the time of treatment selection. This could be more fully acknowledged in the framing of the manuscript.
- While the authors do acknowledge the exploratory nature of the analyses, they could also acknowledge that there tests of significance did not account for multiple comparisons, and thus and significant findings should be interpreted very cautiously.
Author Response
We thank the Reviewer for the favorable appraisal of our manuscript and the helpful comments.
Comment 1: In the background section, the authors highlight literature showing social support could help mitigate PTSD symptoms after ACS. They could also consider literature that shows that some partners may unintentionally worsen PTSD symptoms in ACS patients by transmitting their distress onto their loved ones. (e.g., Cornelius et al. Gen Hosp Psych 2019).
Response: Thank you very much for drawing our attention on this important issue. We added the following sentence to the introduction (Lines 69-71): Nevertheless, it should also be mentioned that significant others may unintentionally worsen PTSD symptoms in patients with ACS by transmitting their distress onto their loved ones [15].
Comment 2: The authors utilize cardiac rehabilitation as an example of health care resources. Yet, attendance at cardiac rehabilitation is often driven as much by patient refusal or drop out rather than resource access. Was lack of attendance or lower number of cardiac rehabilitation visits ever driven by insurance/access issues that were outside the patients’ control? Further, while interventionists could potentially select interventions based on differing amounts of social support, it would be challenging to select based on cardiac rehabilitation or psychotherapy attendance as this would not be known at the time of treatment selection. This could be more fully acknowledged in the framing of the manuscript.
Response: These are excellent points which we added the limitation paragraph in the Discussion (Lines 420-426):
It should be noted that even in a health care system that provides access to CR and psychotherapy on the basis of compulsory health insurance, as is the case in Switzerland, these resources can only be beneficial if patients with ACS are willing to use them. In addition, while interventionists could potentially select interventions based on differing amounts of social support, it would be challenging to select based on CR or psychotherapy attendance as this would not be known for certain at the time of treatment selection.
Comment 3: While the authors do acknowledge the exploratory nature of the analyses, they could also acknowledge that there tests of significance did not account for multiple comparisons, and thus and significant findings should be interpreted very cautiously.
Response: We agree with the Reviewer and added the following sentence to the limitations (Lines 413-415):
Particularly, we did not account our tests of significance for multiple comparisons, so the significant findings should be interpreted very cautiously.
Please see the attachment for our responses to Reveiwer comments.

Reviewer 2 Report
A very interesting article in the field where more research work is needed. PTSD after ACS is nowadays seen and even more recognized among patients on cardiac rehabilitation. Further research will raise clinicians awareness about this ongoing problem and hopefully provide guidelines for better treatment of these patients.
1. This manuscript gives insight into the factors that mediate an early intervention effect (TFC and SFC) in ACS patients in order to prevent the development of PTSD. This is a randomized clinical trial that enrolled 129 patients, includes a psychiatrist and as such gives new findings in this insufficiently researched area of interventions regarding PTSP after ACS. The conclusion of the study is that high social support and cardiac rehabilitation may influence the effect of TFC which is important for the clinicians in order to provide better support. It is important to notice that studied population were mostly men.
2. The manuscript emphasizes the importance of social support and rehabilitation in early intervention with a trauma-focused approach to prevent the development of PTSD symptoms after ACS. This is an important subject for clinicians. However, psychotherapy and cardiac rehabilitation are considered as two different options of support and patients gave information whether they used psychotherapy since hospital discharge and/or cardiac rehabilitation (duration, inpatient, outpatient). Psychotherapy treatment is considered separately from cardiac rehabilitation, but psychological support and overall approach should be/is given to the patients attending cardiac rehabilitation (team of cardiologist, psychiatrist and/or psychologist).
3. Authors gave information that ''some patients also use psychotherapy during CR, however, the effect of CR on PTSD symptoms in the TFC group was independent of psychotherapy attendance'' - only considering patients who used psychotherapy during CR and/or also separately?
The Author concluded ''that other components of CR allowing, for example, correction of cardiac misconceptions, must have been particularly effective in the TFC group. To determine exactly which elements of CR help patients who received early TFC most to manage trauma-related issues, a mixed methods study combining quantitative and qualitative data could be informative.''
The support of a psychologist or a psychiatrist and cardiologist during rehabilitation are important and such may have influence (despite of TFC) on these group of patients and needs further investigation.
Author Response
We thank the Reviewer for the favorable appraisal of our manuscript and the helpful comments.
Comment 1: It is important to notice that studied population were mostly men.
Response: We fully agree. We added the following sentence to the limitation paragraph in the discussion (Lines 417-419):
Because 85% of our study participants were men, it is unclear whether the results would hold true in a sample that was more gender balanced or predominantly female.
Comment 2: Psychotherapy and cardiac rehabilitation are considered as two different options of support and patients gave information whether they used psychotherapy since hospital discharge and/or cardiac rehabilitation (duration, inpatient, outpatient). Psychotherapy treatment is considered separately from cardiac rehabilitation, but psychological support and overall approach should be/is given to the patients attending cardiac rehabilitation (team of cardiologist, psychiatrist and/or psychologist).
Response: We agree and added the following statement to the discussion (Lines 394-395):
However, psychological support for patients attending CR is also provided by the team of cardiologists, psychiatrists, and/or psychologists.
Comment 3: Authors gave information that ''some patients also use psychotherapy during CR, however, the effect of CR on PTSD symptoms in the TFC group was independent of psychotherapy attendance'' - only considering patients who used psychotherapy during CR and/or also separately?
Response: We clarified this issue by adding the following information to the discussion (Lines 392-394):
Unfortunately, we were not able to distinguish whether patients received psychotherapy in the CR setting or outside the CR setting, e.g., in a private psychotherapy practice.
Comment 4: The Authors concluded ''that other components of CR allowing, for example, correction of cardiac misconceptions, must have been particularly effective in the TFC group. To determine exactly which elements of CR help patients who received early TFC most to manage trauma-related issues, a mixed methods study combining quantitative and qualitative data could be informative.'' The support of a psychologist or a psychiatrist and cardiologist during rehabilitation are important and such may have influence (despite of TFC) on these group of patients and needs further investigation.
Response: Thank you for this statement, which we added to the discussion (Lines 398-400):
Clearly, support from psychologists/psychiatrists and cardiologists during CR is important and may have an impact (despite TFC) in patients who have experienced an ACS, which effect needs further investigation.
Please see the attachment for our responses to Reveiwer comments.
